# A Semi-Supervised Abdominal Multi-Organ Pan-Cancer Segmentation Framework with Knowledge Distillation and Multi-Label Fusion

Zengmin Zhang[1,2][0009−0000−4181−0403], Xiaomeng Duan[1][0009−0006−0322−3586], Yanjun Peng[1,2(✉)][0000−0002−8444−0622], and Zhengyu Li[1,2][0009−0007−9420−7932]

[1] College of Computer Science and Engineering, Shandong University of Science and Technology, No. 579, Qianwan'gang Road, Qingdao, 266590, China
[2] Shandong Province Key Laboratory of Wisdom Mining Information Technology, No. 579, Qianwan'gang Road, Qingdao, 266590, China
`{pengyanjuncn}@163.com`

**Abstract.** The segmentation of abdominal organs and tumors plays a crucial role in computer-aided diagnosis of medical images. To achieve high-precision segmentation while maintaining efficiency, especially in semi-supervised learning, we propose a novel semi-supervised knowledge distillation framework. The framework consists of the teacher model and the student model. In the first step, we design an attention nnU-Net with a dual convolutional attention decoder as the teacher model to generate high-quality tumor pseudo-labels for unlabeled tumor data. The dual attention decoder enhances attention to the regions of interest and highlights the most relevant channels, improving the model's ability to optimize features. Additionally, we design an effective 2D sliding window inference strategy to accelerate the inference speed of the teacher model. We utilize partial labels, organ pseudo-labels provided by the FLARE2022 winner, and tumor pseudo-labels for multi-label fusion, ensuring the fusion results closely resemble the ground truth. In the second step, we employ a lightweight nnU-Net as the student model to achieve efficient segmentation. Our method achieved an average DSC score of 88.53% and 30.47% for the organs and lesions on the validation set and the average running time and area under GPU memory-time cure are 15.85s and 15601MB, respectively. Our code is available at https://github.com/zzm3zz/FLARE2023.

**Keywords:** Knowledge Distillation · Dual Attention · Multiple labels fusion · Semi-supervised learning.

## 1 Introduction

Abdominal organs are commonly affected by cancer, such as colorectal cancer and pancreatic cancer, which rank as the second and third leading causes of cancer-related deaths [20]. Therefore, it is necessary to accurately depict abdominal organs and cancerous lesions. However, manual annotation of organs from

CT scans is time-consuming and subjective. As a result, obtaining a large number of fully annotated cases is often impractical. In recent years, deep learning models have achieved state-of-the-art performance in multi-organ or abdominal organ and tumor segmentation tasks, and semi-supervised knowledge distillation has emerged as an important solution to address this issue. FLARE2023 is a competition aimed at efficiently segmenting 13 abdominal organs and pan-cancer lesions in large-scale CT images. In addition to evaluating the accuracy of organ and tumor segmentation, efficiency metrics such as inference time and resource utilization are also taken into consideration. Compared to FLARE2022, FLARE2023 faces the greater challenge of simultaneously segmenting 13 abdominal organs while addressing various lesion tasks associated with abdominal cancers. Furthermore, it explores how to improve segmentation performance using only partially labeled and unlabeled data while maintaining efficient inference.

In recent years, significant efforts have been devoted to exploring image segmentation with partially labeled and unlabeled data. For partially labeled image tasks, a straightforward strategy is to train separate networks on each partially labeled dataset, but this approach leads to longer inference times and higher complexity in post-processing. Recent research has focused on training a single unified model using multiple partially labeled datasets. Zhou et al. [25] proposed the Prior-aware Neural Network (PaNN), which utilizes prior anatomical knowledge of organ sizes estimated from fully labeled datasets to regularize organ distributions in partially labeled datasets. However, this approach requires at least one fully annotated dataset and may not generalize well. Some studies have attempted to design adaptive loss functions that can be directly applied to partially labeled data [3], [19]. Fang et al. [3] introduced the Target Adaptive Loss (TAL), treating unlabeled organs as background. Furthermore, some works have explored the use of training with pseudo-labels, which is also applicable to unlabeled image tasks. Liu et al. [12] proposed training individual models on each partially labeled dataset to generate pseudo-labels for unlabeled organs, followed by supervised training using a pseudo multi-organ dataset. Feng et al. [4] introduced a Knowledge Distillation (MS-KD) framework where a pre-trained teacher model on each partially labeled dataset generates soft pseudo-labels.

We summarize the mainstream approaches in recent years and propose a novel semi-supervised self-training knowledge distillation training framework. It achieves comprehensive segmentation of all organs and lesions while maintaining efficiency. Specifically, we first design a teacher model attention nnU-Net with a dual convolution attention decoder to provide pseudo-labels for tumors. The spatial attention module utilizes dual-path gating to enhance attention to regions of interest, particularly challenging pan-cancerous tumors. The channel attention module adaptively calibrates the connections between low-level and high-level features, emphasizing the most relevant feature channels [7]. Additionally, to accelerate the generation of pseudo-labels for tumors, which may lack annotations in a large number of samples, we propose an effective 2D sliding window inference strategy to speed up the teacher model's inference. Furthermore, considering that tumors may be occluded by genuine organ annotations and that pseudo-labels

from organs may overlap with genuine organ annotations, we design two methods for pseudo-label fusion for partially labeled datasets and unlabeled datasets. Finally, considering efficiency, we employ the small nnU-Net proposed by [10] as the final student model for efficient inference and segmentation. Our main contributions are summarized as follows:

– We propose a novel knowledge distillation training framework, which enables high-precision segmentation and maintains efficiency in a semi-supervised setting.
– We design a teacher model attention nnU-Net that incorporates a dual-convolution attention decoder to achieve high-quality segmentation of regions of interest.
– We propose an effective 2D sliding window inference strategy using prior knowledge of abdominal organ slices to significantly enhance the inference speed of the 2D nnU-Net framework.
– We have devised two label fusion methods to address the issues of inaccuracy and overlap in multi-label scenarios.

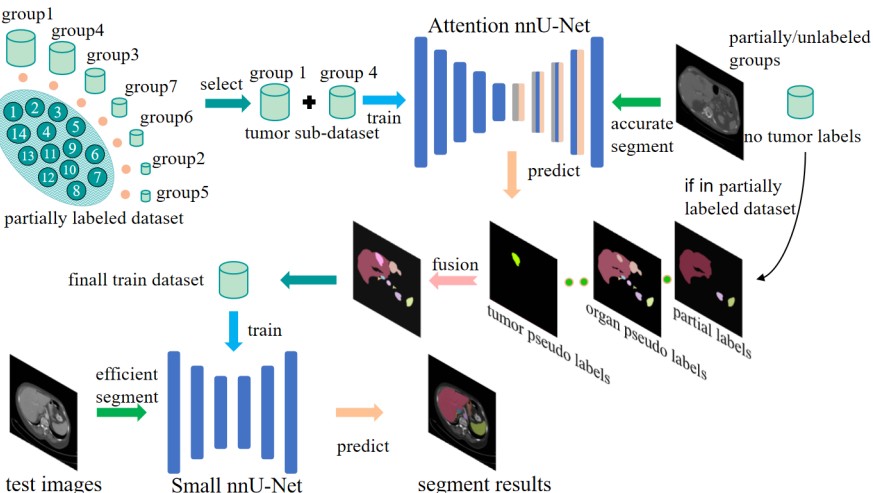

**Fig. 1.** Overview of our proposed framework. It comprises partial label data selection, teacher model attention nnU-Net for tumor segmentation, multi-label fusion methods, and student model small nnU-Net for efficient segmentation.

## 2   Method

The semi-supervised knowledge distillation framework is shown in Figure 1. We divided the partially labeled data of 2,200 cases into 7 groups to represent different combinations of organ or tumor labels based on our careful analysis.

We employ pseudo-labels provided by the highest-precision method [22] from FLARE2022 to label the images that miss organ annotations. To address missing tumor annotations, we train the teacher model attention nnU-Net using group 1 and group 4 data with tumor labels to generate tumor pseudo-labels for the remaining samples in the partially labeled dataset and unlabeled dataset. Section 2.1 elaborates on the methodology that fuses ground truth partial labels, tumors pseudo-labels, and organs pseudo-labels. Finally, we train a small nnU-Net model by use of the final datasets with 4,000 samples to achieve high accuracy and efficiency in the segmentation process.

## 2.1   Preprocessing

The first 2,200 data samples contain partial labels of organs and lesions. Hence, we categorize the data based on the annotation distribution of segmentation targets to train the labeled tumor data. The dataset partitioning results are shown in Table 1.

**Table 1.**  Results of partitioning the partially labeled dataset based on targets annotations. The abbreviations used in Table 1 are as follows: RK, Spl, Pan, Aor, IVC, RAG, LAG, Gall, Eso, Sto, Duo, LK are short for Right Kidney, Spleen, Pancreas, Aorta, Inferior Vena Cava, Right Adrenal Gland, Left Adrenal Gland, Gallbladder, Esophagus, Stomach, Duodenum, Left Kidney.

| Group | Total | Liver | RK | Spl | Pan | Aor | IVC | RAG | LAG | Gall | Eso | Sto | Duo | LK | Tumor |
|---|---|---|---|---|---|---|---|---|---|---|---|---|---|---|---|
| group1 | 888 | | | | | | | | | | | | | | ✓ |
| group2 | 6 | ✓ | ✓ | ✓ | ✓ | | | | | | | | | | |
| group3 | 447 | ✓ | ✓ | ✓ | ✓ | | | | | | | | | ✓ | |
| group4 | 609 | ✓ | ✓ | ✓ | ✓ | | | | | | | | | ✓ | ✓ |
| group5 | 4 | ✓ | ✓ | ✓ | ✓ | ✓ | ✓ | ✓ | | ✓ | ✓ | ✓ | ✓ | ✓ | |
| group6 | 24 | ✓ | ✓ | ✓ | ✓ | ✓ | ✓ | ✓ | ✓ | | ✓ | ✓ | ✓ | ✓ | |
| group7 | 222 | ✓ | ✓ | ✓ | ✓ | ✓ | ✓ | ✓ | ✓ | ✓ | ✓ | ✓ | ✓ | ✓ | |

We perform image reorientation to align the images with the target orientation for each modality-specific data. The resource-intensive teacher model is expected to generate high-quality tumor pseudo-labels, which inevitably leads to higher resource consumption. Therefore, we adopted a 2D method with a smaller memory footprint. A more effective 3D method is employed for segmentation in the lightweight student model. Our configuration information is provided in Table 2. Intensity normalization is conducted using the default method of nnU-Net [11]. We set intensity values below 14 to 0 in the ground truth of group 1 and group 4, and intensity values of 14 to 1 to improve the accuracy of tumor segmentation in the teacher model. This conversion transforms the task of multi-organ tumor segmentation into a binary tumor classification. Subsequently, the labeling intensity is adjusted during the label fusion process.

**Table 2.** Comparison of different strategies. The first one is the default 3D nnU-Net configuration. The input patch sizes and inter-axis spacing are denoted as (z, y, x) or (y, x).

| Setting | Default | Attention nnU-Net | Small nnU-Net |
|---|---|---|---|
| method | 3D | 2D | 3D |
| channels in the first stage | 32 | 32 | 16 |
| convolution number per stage | 2 | 2 | 2 |
| downsampling times | 5 | 7 | 4 |
| input patch size | (40, 224, 192) | (512, 512) | (32, 128, 192) |
| input spacing | (2.5, 0.8, 0.8) | (0.8164, 0.8164) | (4, 1.2, 1.2) |
| test time augmentation | yes | no | no |

## 2.2 Proposed Method

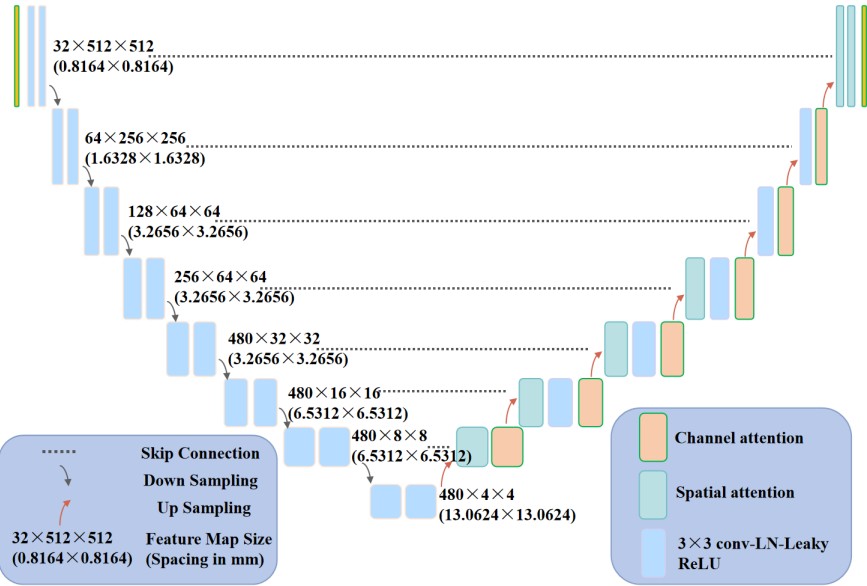

**Fig. 2.** The network architecture of Attention nnU-Net

**Resource Intensive Attention nnU-Net.** We design attention nnU-Net to label tumor pseudo-labels more effectively. As shown in Figure 2, our attention nnU-Net expands upon the 2D nnU-Net architecture by incorporating a dual attention mechanism comprised of spatial attention (SA) and channel attention (CA) in the decoder [7]. The spatial attention utilizes dual-path gating to enhance attention towards regions of interest, particularly challenging pan-cancer tumors. The channel attention module dynamically adjusts the connections between low-level and high-level features, enabling higher coefficients to

be assigned to more relevant channels, thereby emphasizing the most pertinent feature channels.

Our SA block is a type of dual-path spatial attention that utilizes two attention gates simultaneously to enhance attention to regions of interest and reduce noise in the attention maps. Figure 3(a) illustrates the detailed information of a single attention gate pathway. Here, $z^l$ represents the low-level feature map in the encoder, while $z^h$ represents the high-level feature map upsampled from the decoder. Both $z^h$ and $z^l$ are compressed with 1×1 convolutions, with the output channels $C$, and then summed and passed through a $ReLU$ activation function. The activated feature map is then fed into another 1×1 convolution with one output channel, and the resulting attention coefficients $\alpha \in [0,1]^{H \times W}$ on a pixel level are obtained through the $Sigmoid$ function. Subsequently, $z^l$ is calibrated by multiplication with $\alpha$. In the dual-path attention gate, the spatial attention maps in the two pathways are represented individually as $\hat{\alpha}$ and $\widetilde{\alpha}$, as shown in Figure 3(b), and the output of the dual-channel attention gate is obtained as:

$$Output = ReLU\left[\varphi^C((z^l \cdot \hat{\alpha})\copyright(z^l \cdot \widetilde{\alpha}))\right] \tag{1}$$

where $\copyright$ indicates channel concatenation and $\varphi^C$ represents a 1×1 convolution with $C$ output channels.

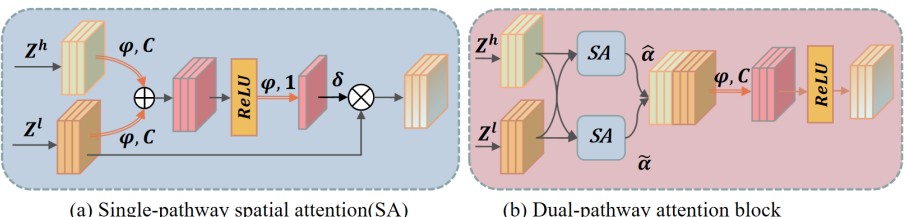

(a) Single-pathway spatial attention(SA)          (b) Dual-pathway attention block

**Fig. 3.** Details of our spatial attention(SA).

To better utilize the most informative feature channels, we introduce channel attention to automatically emphasize relevant feature channels while suppressing irrelevant ones. Our channel attention (CA) block primarily combines the low-level features from the encoder, calibrated by spatial attention, and the high-level features from the decoder, as shown in Figure 4. Let $z$ represent the input features with $C$ channels. We use global average pooling $P_{avg}$ and global max pooling $P_{max}$ to obtain the global information for each channel, resulting in $P_{avg}(x) \in \mathbb{R}^{C \times 1 \times 1}$ and $P_{max}(x) \in \mathbb{R}^{C \times 1 \times 1}$, respectively. The channel attention coefficients $\beta$ are obtained using a multilayer perceptron ($MLP$) and can be represented as $\beta \in [0,1]^{C \times 1 \times 1}$. The results are summed and input to $Sigmoid$ to obtain $\beta$, and the output can be represented as:

$$Output = z \cdot \beta + z \tag{2}$$

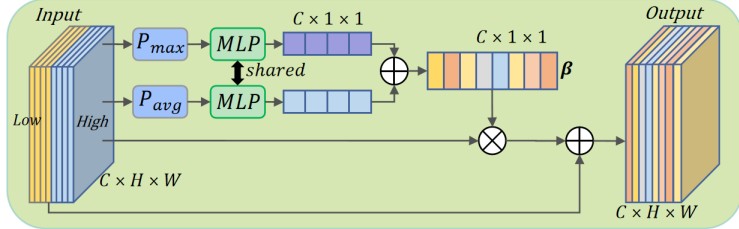

**Fig. 4.** Details of our channels attention(CA).

We use the summation between Dice loss and cross-entropy loss because compound loss functions have been proven to be robust in various medical image segmentation tasks [13].

**Fusion strategy for partial labels.** The simple fusion method performs direct fusion of partial labels, organs, and tumor pseudo-labels, and chooses to use the value of the partial label if the positions overlap when the annotation is not zero. However, the simple fusion method ignores the inaccuracy of organ pseudo-label edges and the fact that tumors can be masked by organ labeling. We therefore perform dual fusion on partial labels to obtain a more comprehensive and reasonable distribution of target annotations. Specifically, for group $i$, where $i \in \{2, 3, 5, 6, 7\}$, there may be missing tumor annotations. We incorporate high-quality tumor pseudo-labels generated by the teacher model for fusion. As real tumors can be masked by organ annotations, when there is an overlap between organ annotations and tumor pseudo-annotations, we prioritize the tumor pseudo-annotations. The specific method is described in Equation 3, where $\theta_p$ represents the ground truth partial annotations, and $\theta_t$ represents the tumor pseudo-annotations.

$$\hat{\theta}^{(z,x,y)} = \begin{cases} \theta_p^{(z,x,y)}, \; \theta_t^{(z,x,y)} = 0 \\ \theta_t^{(z,x,y)}, \; \theta_t^{(z,x,y)} \neq 0 \end{cases} \tag{3}$$

Subsequently, for group $j$ where $j \in \{1, 2, 3, 4, 5, 6\}$ with remaining missing organ annotations, we fuse $\hat{\theta}^{(z,x,y)}$ with corresponding organ pseudo-labels [22] using the approach outlined in Equation 4. $\theta_o$ represents organ pseudo-labels. $\lambda$ represents the annotated organ categories in group $j$.

$$\theta^{(z,x,y)} = \begin{cases} \hat{\theta}^{(z,x,y)}, \; \hat{\theta}^{(z,x,y)} \neq 0 \\ \theta_o^{(z,x,y)}, \; \hat{\theta}^{(z,x,y)} = 0 \wedge \theta_o^{(z,x,y)} \neq \lambda \\ 0 \qquad otherwise \end{cases} \tag{4}$$

Different from the tumor pseudo-label fusion strategy, there are two scenarios for organ pseudo-label fusion. (1)When the organ pseudo-annotations overlap with the foreground of the $\hat{\theta}^{(z,x,y)}$, we use the annotations of $\hat{\theta}^{(z,x,y)}$ in overlapping regions. (2) In the background region of $\hat{\theta}^{(z,x,y)}$, if the corresponding

position in the organ pseudo-label is annotated as a missing organ in $\hat{\theta}^{(z,x,y)}$, we use the organ annotations labeled by the organ pseudo-label. However, if the corresponding position is annotated as background or as an already annotated organ in $\hat{\theta}^{(z,x,y)}$, it is considered as background.

**Fusion strategy for unlabeled images pseudo-labels.** For unlabeled images, similar to the partial label fusion strategy, in cases of overlap at the same location, we give priority to the tumor pseudo-label. The fusion method is described in Equation 5.

$$\theta^{(z,x,y)} = \begin{cases} \theta_o^{(z,x,y)}, \theta_t^{(z,x,y)} = 0 \\ \theta_t^{(z,x,y)}, \theta_t^{(z,x,y)} \neq 0 \end{cases} \tag{5}$$

**Efficient 2D Sliding Window Inference.** We employed a precise configuration and trained using a large-scale 2D attention nnU-Net to infer tumors. This inevitably leads to higher resource consumption and longer inference time. Based on [10], we have designed an efficient 2D sliding window inference method. The default sliding window strategy is designed with separate steps for the X and Y axes, using a nested two-layer for loop to iterate over the image. However, a significant portion of 2D slices in the entire 3D image does not contain the abdominal organs and tumor regions, especially in whole-body CT images. Moreover, the abdominal organs and tumors should be located in the middle of the slice plane. Therefore, we use $2 \times 2$ windows on each cross-sectional plane, with a window size of $512 \times 512$. As shown in Figure 5, since the resized 2D slices have similar dimensions to the window size after resampling, we start by performing inference on the first window. We can skip the remaining three windows if this window does not contain any foreground regions.

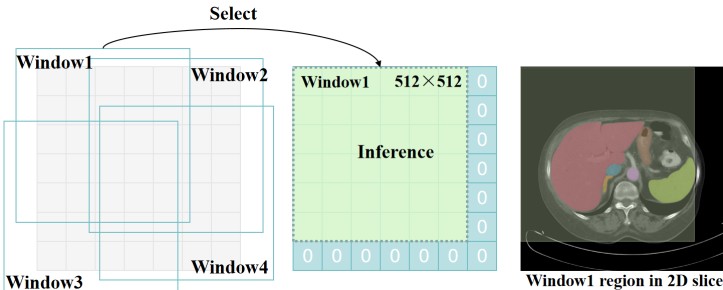

**Fig. 5.** On the left is the default inference strategy of 2D nnU-Net. The middle image illustrates our proposed strategy. The image on the right demonstrates that window 1 covers a large portion of the abdominal area. The inference of the remaining three windows depends on whether window 1 detects the target.

### 2.3    Post-processing

Our method does not utilize post-processing operations because techniques such as connected component analysis and testing time augmentation tend to introduce significant computational overhead during the prediction phase.

## 3    Experiments

### 3.1    Dataset and evaluation measures

The FLARE 2023 challenge is an extension of the FLARE 2021-2022 [15][16], aiming to aim to promote the development of foundation models in abdominal disease analysis. The segmentation targets cover 13 organs and various abdominal lesions. The training dataset is curated from more than 30 medical centers under the license permission, including TCIA [2], LiTS [1], MSD [21], KiTS [8,9], autoPET [6,5], TotalSegmentator [23], and AbdomenCT-1K [17]. The training set includes 4000 abdomen CT scans where 2200 CT scans with partial labels and 1800 CT scans without labels. The validation and testing sets include 100 and 400 CT scans, respectively, which cover various abdominal cancer types, such as liver cancer, kidney cancer, pancreas cancer, colon cancer, gastric cancer, and so on. The organ annotation process used ITK-SNAP [24], nnU-Net [11], and MedSAM [14].

The evaluation metrics encompass two accuracy measures—Dice Similarity Coefficient (DSC) and Normalized Surface Dice (NSD)—alongside two efficiency measures—running time and area under the GPU memory-time curve. These metrics collectively contribute to the ranking computation. Furthermore, the running time and GPU memory consumption are considered within tolerances of 15 seconds and 4 GB, respectively.

### 3.2    Implementation details

**Table 3.** Development environments and requirements.

| System | Ubuntu 20.04.3 LTS |
|---|---|
| CPU | AMD EPYC 7T83@3.50GHz |
| RAM | 1×90GB; 320MT/s |
| GPU (number and type) | 1 NVIDIA RTX 4090 24G |
| CUDA version | 11.3 |
| Programming language | Python 3.8.10 |
| Deep learning framework | torch 1.12.1, torchvision 0.13.1 |
| Specific dependencies | nnunet 1.7.0 |
| Code | https://github.com/zzm3zz/FLARE2023 |

The development environments and requirements are presented in Table 3. The training protocols of attention nnU-Net and small nnU-Net are listed in Table 4. During the training process, we employed data augmentation techniques

such as dynamic additive brightness, gamma adjustment, rotation, scaling, and elastic deformation. Mirror data augmentation is not used in either of the networks since test-time augmentation (TTA) involving flipping is not performed during inference.

After training on group 1 and group 4 and obtaining high-precision pseudo-labels for all potentially missing tumor annotated samples, we conducted four experiments. In the first experiment, we obtained fused labels for the first 2,200 data using a simple label fusion method and conducted training evaluation using the teacher model. In the second experiment, we utilized the proposed label fusion method and continued the evaluation using the teacher model. In the third round, based on the data from the previous round, we conducted an evaluation using the student model. In the fourth round, different from the third experiment, we employed all 4,000 samples and used the student model to conduct an evaluation.

**Table 4.** Training protocols for the refine model.

| Model | Attention nnU-Net / Small nnU-Net |
|---|---|
| Network initialization | "He" normal initialization |
| Batch size | 4 / 2 |
| Patch size | 512×512 / 32×128×192 |
| Total epochs | 1000 / 1500 |
| Optimizer | SGD with nesterov momentum ($\mu = 0.99$) |
| Initial learning rate (lr) | 0.005 / 0.01 |
| Lr decay schedule | Poly learning rate policy: $(1 - \text{epoch}/\text{Total epochs})^{0.9}$ |
| Training time | 19 hours / 17 hours |
| Number of model parameters | 62.96M / 5.64M |
| Number of flops | 114G / 70G |

## 4   Results and discussion

### 4.1   Quantitative results on validation set

Table 5 shows our final submission result. We trained 4,000 cases of train data by using the student model small nnU-Net with the proposed label fusion method and achieved an average DSC score of 84.83% and an average NSD score 89.60% on the public validation set. Similarly, on the online validation set, we achieved an average DSC score of 84.38% and an average NSD score 89.28%.

Table 6 shows the results of our ablation experiments on the online validation set, corresponding to the four experiments mentioned above. Comparing the first and second columns, our label fusion method improved the segmentation accuracy of tumors by 9.62%. When comparing the second and third columns,

**Table 5.** Quantitative evaluation results of our final submitted model.

| Target | Public Validation | | Online Validation | | Testing | |
|---|---|---|---|---|---|---|
| | DSC(%) | NSD(%) | DSC(%) | NSD(%) | DSC(%) | NSD (%) |
| Liver | 97.12 ± 2.38 | 98.25 ± 4.45 | 97.06 | 98.24 | 96.09 | 96.93 |
| Right Kidney | 93.83 ± 7.25 | 94.64 ± 9.05 | 93.07 | 94.06 | 94.06 | 94.52 |
| Spleen | 96.34 ± 1.41 | 97.92 ± 3.73 | 95.38 | 97.16 | 95.90 | 97.94 |
| Pancreas | 85.19 ± 6.36 | 96.36 ± 5.86 | 83.24 | 95.10 | 87.77 | 96.66 |
| Aorta | 95.36 ± 2.53 | 98.72 ± 2.76 | 95.40 | 98.63 | 96.04 | 99.60 |
| Inferior vena cava | 91.01 ± 4.93 | 93.12 ± 5.43 | 90.80 | 92.69 | 91.96 | 94.32 |
| Right adrenal gland | 81.22 ± 16.9 | 93.15 ± 19.1 | 82.46 | 94.91 | 83.58 | 96.00 |
| Left adrenal gland | 83.20 ± 5.78 | 96.02 ± 3.27 | 82.59 | 94.96 | 84.20 | 96.18 |
| Gallbladder | 83.22 ± 23.6 | 84.40 ± 24.7 | 82.56 | 83.54 | 80.93 | 83.29 |
| Esophagus | 80.57 ± 15.9 | 91.16 ± 16.4 | 81.55 | 92.59 | 87.29 | 97.19 |
| Stomach | 92.88 ± 4.65 | 96.43 ± 5.38 | 93.26 | 96.83 | 93.25 | 96.77 |
| Duodenum | 81.55 ± 8.25 | 94.55 ± 5.46 | 81.53 | 94.44 | 85.18 | 96.04 |
| Left kidney | 91.73 ± 14.3 | 92.52 ± 15.7 | 92.02 | 93.07 | 92.96 | 93.95 |
| Tumor | 34.46 ± 34.6 | 27.21 ± 29.0 | 30.47 | 23.67 | 36.35 | 23.96 |
| Average Organs | 88.71 | 94.40 | 88.53 | 94.32 | 89.77 | 95.23 |
| Average All | 84.83 | 89.60 | 84.38 | 89.28 | 85.95 | 90.14 |

**Table 6.** Ablation experimental results on online validation leaderboard. Data denotes the number of training samples we used. Labels denote the way we fused the labels. The first experiment uses simple fusion. The last three times used our proposed method.

| Method | Att nnU-Net | | Att nnU-Net | | Small nnU-Net | | Small nnU-Net | |
|---|---|---|---|---|---|---|---|---|
| Data | 2,200 | | 2,200 | | 2,200 | | 4,000 | |
| Labels | Simple Fusion | | Our Method | | Our Method | | Our Method | |
| Target | DSC(%) | NSD(%) | DSC(%) | NSD(%) | DSC(%) | NSD(%) | DSC(%) | NSD(%) |
| Liver | 98.25 | 98.14 | 98.27 | 97.49 | 96.58 | 97.42 | 97.06 | 98.24 |
| RK | 92.30 | 92.12 | 92.30 | 91.54 | 89.58 | 88.98 | 93.07 | 94.06 |
| Spleen | 95.38 | 94.85 | 98.00 | 97.33 | 95.07 | 96.69 | 95.38 | 97.16 |
| Pancreas | 85.08 | 94.61 | 85.76 | 94.80 | 81.50 | 94.27 | 83.24 | 95.10 |
| Aorta | 97.24 | 98.94 | 96.53 | 98.31 | 95.25 | 98.35 | 95.40 | 98.63 |
| IVC | 90.58 | 90.88 | 89.77 | 89.61 | 89.24 | 90.60 | 90.80 | 92.69 |
| RAG | 87.80 | 96.59 | 86.54 | 94.99 | 79.86 | 92.62 | 82.46 | 94.91 |
| LAG | 84.74 | 92.83 | 83.75 | 91.58 | 79.95 | 92.71 | 82.59 | 94.96 |
| Gallbladder | 87.37 | 87.82 | 84.93 | 85.01 | 79.54 | 79.39 | 82.56 | 83.54 |
| Esophagus | 83.68 | 93.48 | 83.99 | 93.64 | 79.98 | 91.69 | 81.55 | 92.59 |
| Stomach | 93.79 | 95.49 | 93.34 | 94.66 | 91.96 | 95.35 | 93.26 | 96.83 |
| Duodenum | 82.72 | 94.37 | 80.59 | 93.28 | 77.32 | 92.57 | 81.53 | 94.44 |
| LK | 90.73 | 90.56 | 92.64 | 91.98 | 91.40 | 91.14 | 92.02 | 93.07 |
| Tumor | 25.18 | 19.53 | 34.80 | 26.01 | 24.81 | 16.89 | 30.47 | 23.67 |
| Avg Organs | 88.97 | 93.90 | 89.72 | 93.40 | 86.71 | 92.44 | 88.53 | 94.32 |
| Average All | 85.34 | 88.58 | 85.80 | 88.58 | 82.29 | 87.04 | 84.38 | 89.28 |

the attention nnU-Net teacher model outperformed the small nnU-Net student model with higher scores of 3.51% and 1.54% in DSC and NSD, respectively. Additionally, comparing the third and fourth columns, including an additional 1,800 cases resulted in a 2.09% and 2.24% improvement in DSC and NSD, respectively.

**Table 7.** Ablation for tumor segmentation on public validation set

| Method | Tumor DSC(%) | Tumor NSD(%) |
|---|---|---|
| 2D nnU-Net | 34.28 | 27.29 |
| Attention nnU-Net | 36.75 | 29.51 |

Table 7 demonstrates the ablation experiments for tumor segmentation ability. We use group1 and group4 contained 1497 cases with tumor label to evaluate the ability of tumor segmentation. It can be seen that the addition of the dual convolution attention decoder improves the model's ability to segment tumors compared to the regular 2D nnU-Net resulting in a 2.47% and 2.22% improvement in DSC and NSD, respectively.

We compared the effectiveness of the model and sample selection. It can be observed that using a larger number of samples improves the segmentation results of organs and lesions. The utilization of resource-intensive models yields better segmentation performance compared to smaller models by using more parameters and calculations. Additionally, there is significant room for improvement in lesion segmentation. After balancing accuracy and performance, we ultimately selected the student model with 4,000 train cases that exhibited the most comprehensive performance as our submission result.

### 4.2   Qualitative results on validation set

Figure 6 shows our final four representative segmentation results on the validation set. Additionally, we present the segmentation results using small nnU-Net training 2,200 partially labeled data for comparison. For Case #43 and Case #81, our network achieved high-precision recognition of all organs and lesions. However, for Case #35, it is evident that our model exhibits segmentation defects for tumors and struggles to correctly identify the targets in cases with multiple overlapping objects. Additionally, for Case #67, we found that our model has difficulty recognizing the esophagus.

We believe that there are two factors contributing to the suboptimal segmentation results. Firstly, for small organs and lesions, the target regions are small, exhibit significant shape deformations, and have low contrast and unclear boundaries. In particular, as lesions can occur within different organs, accurately determining their precise locations becomes challenging. Some lesions only occupy a small portion of the entire sample, making it difficult for our model to differentiate them. Additionally, we attribute these issues to the inaccuracies in

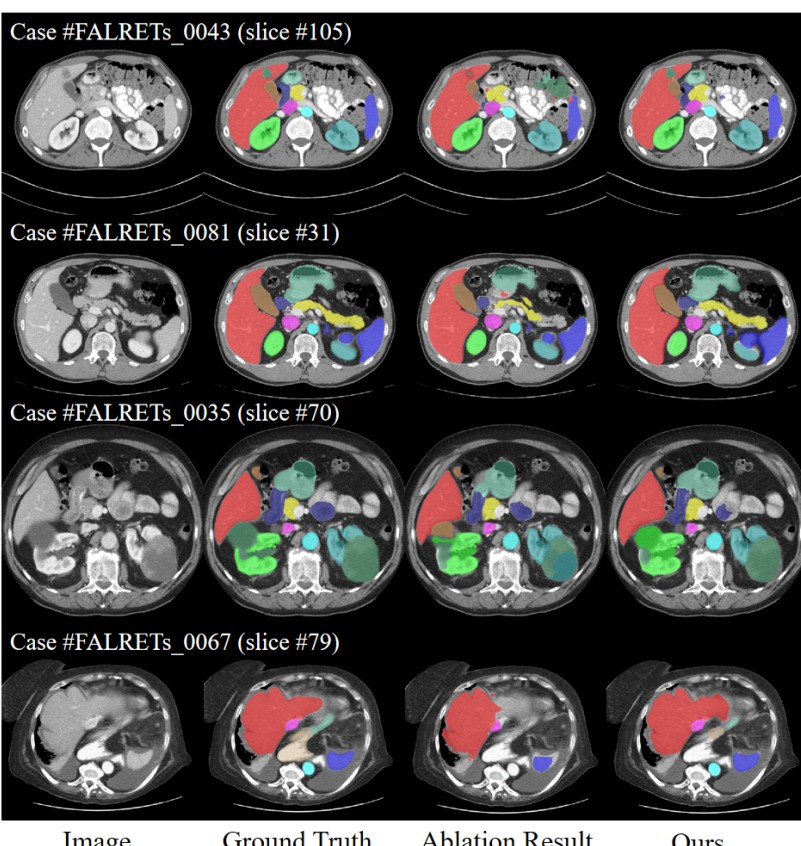

**Fig. 6.** Qualitative results of our small nnUNet and ablation comparative experiment on two easy cases (Case #43 and Case #81) and two hard cases (Case #35 and Case #67)

the mixed annotations labels, as well as the loss of important details due to the lower resolution resulting from image resampling.

### 4.3   Segmentation efficiency results on validation set

Table 8 presents the evaluation results of our validation dataset Docker submission, including all running time, max GPU memory usage, and total GPU memory.

**Table 8.** Quantitative evaluation of segmentation efficiency in terms of the running them and GPU memory consumption. Total GPU denotes the area under GPU Memory-Time curve. Evaluation GPU platform: NVIDIA QUADRO RTX5000 (16G).

| Case ID | Image Size | Running Time (s) | Max GPU (MB) | Total GPU (MB) |
|---|---|---|---|---|
| 0001 | (512, 512, 55) | 18.21 | 2562 | 16305 |
| 0051 | (512, 512, 100) | 15.03 | 1694 | 15786 |
| 0017 | (512, 512, 150) | 26.23 | 3704 | 21449 |
| 0019 | (512, 512, 215) | 14.21 | 2562 | 14479 |
| 0099 | (512, 512, 334) | 15.20 | 1978 | 15303 |
| 0063 | (512, 512, 448) | 17.86 | 1694 | 18322 |
| 0048 | (512, 512, 499) | 19.55 | 1694 | 20940 |
| 0029 | (512, 512, 554) | 21.16 | 3704 | 22890 |

### 4.4   Results on final testing set

We submitted the docker of our final solution and was evaluated by the Flare official. The results on final testing set are shown in Table 5.

### 4.5   Limitation and future work

Due to the large number of data samples and limited experimental resources and time, the quality of pseudo-labels is not satisfactory, resulting in suboptimal results. In future work, we will reference the latest research advancements to improve efficiency and enhance segmentation quality.

## 5   Conclusion

In this paper, we followed a semi-supervised knowledge distillation strategy and proposed a novel semi-supervised knowledge distillation framework. This framework consists of the teacher model and the student model. In the first step, we design a resource-intensive teacher model, attention nnU-Net, which incorporates a dual convolutional attention decoder, to generate accurate tumor pseudo-labels. Additionally, we designed an effective 2D sliding window inference strategy to

accelerate pseudo-label generation. Subsequently, we devised a method for multi-label fusion to enhance target segmentation accuracy. In the second step, we employ a lightweight nnU-Net as the student model to achieve efficient segmentation. Experimental results on the FLARE2023 validation set demonstrated that our method exhibits excellent segmentation performance and efficiency. In the future, we will continue to optimize the framework to further improve the model's segmentation performance and enable fast, low-resource inference.

**Acknowledgements** The authors of this paper state that the methods implemented for participating in the FLARE2023 challenge do not utilize any pretrained models, nor do they involve data augmentation or additional manual annotations. The proposed approach is fully automated and does not require any human intervention. We thank all the data owners for making the CT scans publicly available and CodaLab [18] for hosting the challenge platform.

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

**Table 9.** Checklist Table. Please fill out this checklist table in the answer column.

| Requirements | Answer |
| --- | --- |
| A meaningful title | Yes |
| The number of authors ($\leq 6$) | 4 |
| Author affiliations, Email, and ORCID | Yes |
| Corresponding author is marked | Yes |
| Validation scores are presented in the abstract | Yes |
| Introduction includes at least three parts: background, related work, and motivation | Yes |
| A pipeline/network figure is provided | Figure 1 |
| Pre-processing | Page 4 |
| Strategies to use the partial label | Page 7 |
| Strategies to use the unlabeled images. | Page 8 |
| Strategies to improve model inference | Page 8 |
| Post-processing | Page 9 |
| Dataset and evaluation metric section is presented | Page 9 |
| Environment setting table is provided | Table 3 |
| Training protocol table is provided | Table 4 |
| Ablation study | Table 6, 7 |
| Efficiency evaluation results are provided | Table 8 |
| Visualized segmentation example is provided | Figure 6 |
| Limitation and future work are presented | Yes |
| Reference format is consistent. | Yes |