# OpenReview forum: "A Semi-Supervised Abdominal  Multi-Organ Pan-Cancer Segmentation Framework with Knowledge Distillation and Multi-Label Fusion"
_MICCAI.org/2023/FLARE — Submitted to FLARE 2023_

### Official Review · Reviewer_rGkh · 2023-09-19
**A Semi-Supervised Abdominal Multi-Organ Pan-Cancer Segmentation Framework with Knowledge Distillation and Multi-Label Fusion**

**Rating:** 6
**Confidence:** 5

**Review:**

This paper proposes a semi-supervised self-training knowledge distillation training framework, where an attention nnU-Net with dual convolution attention decoder is proposed to train the teacher model. Then the knowledge learned from the teacher branch is distilled to a smaller model for better efficiency. The authors also introduce the  label fusion methods to tackle overlap in multi-label scenarios..
The authors present their ideas clearly in the submitted paper. The paper is overall well-written and easy to follow. The designed framework demonstrates reasonable results on the organs with good efficiency. However,  it is not clear if the dual attention module works well on the tumors. Besides,  from Table 6, results using the proposed label fusion method seems to generate worse performance than the simple fusion strategy. The manuscript needs further proofreading to meet the publication standard.

---

> ### Author Response · Authors · 2023-11-04
> **We have revised manuscript and provide a point-by-point response to each of the issues raised by reviewer rGkh.**
>
> We are very grateful for your great efforts on our paper and thank you again for your valuable comments. Below we will provide a detailed point-by-point response to your concerns.
>
> Comment 1: However, it is not clear if the dual attention module works well on the tumors.
>
> Response 1: We are very sorry that we do not provide an ablation experiment of the dual attention module, which may cause you concern. In the revised manuscript, we provide Table 7 of the comparative experimental results of attention nnU-Net and 2D nnU-Net on the tumor segmentation task to prove the effect of the dual attention module on tumor segmentation.
>
> Comment 2: From Table 6, results using the proposed label fusion method seems to generate worse performance than the simple fusion strategy.
>
> Response 2: Thank you for raising your concerns. Our manuscript does not fully demonstrate the experimental results. In the revised manuscript, we modified the result display of table5 and table6 to make the segmentation results of organs and tumors significantly different. It can be seen that the label fusion method we proposed produced ideal results in tumor segmentation.

---

### Official Review · Reviewer_vQSE · 2023-09-24
**Nice work with high efficiency and beautiful figures**

**Rating:** 8
**Confidence:** 5

**Review:**

Summary:
The paper presents a semi-supervised knowledge distillation framework, featuring a meticulously designed high-precision attention nnU-Net with a dual-convolution attention decoder, alongside an efficient small nnU-Net. Additionally, the paper introduces an effective 2D sliding window inference strategy and two label fusion methods to enhance segmentation efficiency and accuracy.

Pros:
1. The paper is well-written, and the figures are exquisitely drawn.
2. The method yields good quantitative results, demonstrating fast inference speed and low GPU memory consumption.

Cons:
The content of Table 4 should be carefully reviewed, as some information appears to be inconsistent or inaccurate.

---

> ### Author Response · Authors · 2023-11-04
> **We have revised manuscript and provide a point-by-point response to each of the issues raised by reviewer vQSE.**
>
> Thank you very much for your high evaluation of our work and your concerns about the review. We will respond to your questions in detail.
>
> Comment 1: The content of Table 4 should be carefully reviewed, as some information appears to be inconsistent or inaccurate.
>
> Response 1: Thank you very much for raising the issue regarding the inaccuracy of our article. After careful re-calculation, the correct data in Table 4 has been corrected.

---

### Official Review · Reviewer_hXcb · 2023-10-03
**A Semi-Supervised Abdominal Multi-Organ Pan-Cancer Segmentation Framework with Knowledge Distillation and Multi-Label Fusion**

**Rating:** 7
**Confidence:** 5

**Review:**

Strengths:
1. The data set was carefully analyzed and divided in the preprocessing stage, and corresponding training strategies were formulated subsequently.
2. The paper is generally well-written and easy to follow, and the figures are exquisitely drawn.
3. The 2D Sliding Window Inference method proposed in this article can effectively improve segmentation efficiency while ensuring segmentation accuracy.

Weaknesses:
1. Ablation experiments can't prove whether the dual attention module is effective for tumor segmentation.
2. This article does not represent the general process of the simple fusion strategy, so it is impossible to compare the specific differences between it and the strategy proposed by the author.

---

> ### Author Response · Authors · 2023-11-04
> **We have revised manuscript and provide a point-by-point response to each of the issues raised by reviewer hXcb.**
>
> We greatly appreciate your contribution to our manuscript and thank you again for your valuable comments. We have developed a point-by-point response based on your comments to address your concerns about us.
>
> Comment 1:  Ablation experiments can't prove whether the dual attention module is effective for tumor segmentation.
>
> Response 1: We are very sorry that we neglected to provide this ablation experiment. We performed ablation experiments to demonstrate the effect of the dual attention module on tumor segmentation. The results are shown in Table 7. It can be seen that the addition of the dual convolution attention decoder improves the model’s ability to segment tumors compared to the regular 2D nnU-Net resulting in a 2.47% and 2.22% improvement in DSC and NSD, respectively.
>
> Comment 2: This article does not represent the general process of the simple fusion strategy, so it is impossible to compare the specific differences between it and the strategy proposed by the author.
>
> Response 2: Thank you very much for your concerns, we have overlooked this issue. We added a detailed description and disadvantages of the simple fusion method in "Fusion strategy for partial labels" in Section 2.2 to prove the necessity of optimizing the fusion method.

---

### Decision · Program_Chairs · 2023-10-24

Accept